# Male partner involvement in postnatal care service utilization and associated factors: A community-based cross-sectional study in Motta district, Northwest Ethiopia

**Alemwork Abie**[1]*, **Getahun Belay**[1], **Eden Asmare**[1], **Nigusu Ayalew**[1], **Wondu Feyisa**[1], **Nigus Kassie**[2]

1 Department of Midwifery, College of Medicine and Health Science, Bahir Dar University, Bahir Dar, Ethiopia, 2 Department of Reproductive Health, College of Medicine and Health Science, Dire Dawa University, Dire Dawa, Ethiopia

* abiealemwork84@gmail.com

**Data Availability Statement:** All relevant data are within the paper and its Supporting information files.

## Abstract

### Background

Male partner involvement during the postpartum period is an effective strategy to improve maternal and newborn health outcomes. However, since reproductive health has been viewed as only a woman's issue, it remains a major challenge in developing countries, including Ethiopia. The current study aimed to assess male partner involvement in postnatal care service utilization and associated factors in the Motta district of North West Ethiopia in 2020.

### Methods

A community-based cross-sectional study was conducted from March 16 to 30, 2020, among male partners whose wives gave birth in the last twelve months in Motta district. A systematic random sampling technique was used to obtain 612 study participants. Data entry was carried out by using Epi Data version 3.1 and exported to Statistical Package of Social Science version 23 for analysis. A binary and multiple logistic regression method were employed to estimate the crude and adjusted odds ratios with a confidence interval of 95% and a P value of less than 0.05 were considered statistically significant.

### Results

The findings from this study highlight that the overall male partner involvement in postnatal care service utilization was 20.8% at 95%CI (17.6%, 24.1%). The regression results indicated that male partners residing in urban areas, attending primary and secondary education, having good knowledge of postnatal care services, having good knowledge of danger signs and complications during the postnatal period, having a favorable attitude toward male partner involvement in postnatal care, and a short distance to health care facilities were shown to be significantly and positively associated with male partner involvement in postnatal care services.

**Funding:** The study was granted 25,000 ETB by Bahir Dar University College of Medicine and health science, Bahir Dar, Ethiopia and received by the corresponding author. But the funding organization did not have any role in the designing of the study, data collection, analysis and interpretation of data, and writing the manuscript.

**Competing interests:** The authors have declared that no competing interests exist.

## Conclusion

The magnitude of male partner involvement in postnatal care service utilization was low as compared to other studies. Community-based awareness creation through health education and expanding educational and health care service institutions are essential to increase the involvement of male partners in postnatal care services.

## Introduction

Male partner involvement in Maternal, Newborn, and Child Health (MNCH) is described as a development of social and behavioral modification to play more responsible roles in MNCH with the rationale of ensuring women's reproductive health and the well-being of the children and society as well [1, 2]. Male partners participation in maternal health in general and Post Natal Care (PNC) in particular is essential in developing countries like Ethiopia where most women rely on their male partners decisions regarding their health [3]. Many studies revealed the merits of male partner involvement in maternal health; relief of stress, pain, and anxiety during delivery; increased maternal access to Antenatal Care (ANC) and PNC services; improved maternal mental health; and increased contraceptive usage [4–7]. Studies also shown that delays in the decision to seek care and reaching care are directly related to the involvement of male partners because they operate as supportive caretakers, negotiators, facilitators for transportation, and promoters [8, 9]. In addition, they are also the primary decision-makers of many issues in the household, including health-related decisions that influences women and children [10, 11].

The available evidence indicated that approximately 210 million women become pregnant each year with 30 million (15%) developing life-threatening complications resulting in death, of which the majority of maternal deaths are attributed to low male involvement [12]. In general, developing countries accounts more than 99% of all maternal deaths, about half of which occur in sub-Saharan Africa, and the majority (62%) occur during the postpartum period [13]. Similarly, 2.5 million neonates died in the first month of life worldwide in 2017, which is the highest in sub-Saharan Africa and South Asia, with each estimated at 27 deaths per 1,000 live births [14]. According to the 2016 Ethiopian Demographic Health Survey (EDHS), the Maternal Mortality Ratio (MMR) and neonatal mortality in the country were estimated about 412 per 100,000 and 29 per 1,000 live births respectively [15]. Alleviating this serious problem requires great male partners participation in MNCH services, above all in PNC services, because PNC is a key factor in the reduction of maternal and neonatal mortality [16].

In the majority of Sub-Saharan African countries, including Ethiopia, male partners participation in MNCH services, particularly in PNC services is still overlooked [17]. This is primarily because reproductive health has been viewed as only a woman's affair and most reproductive health programs are exclusively focused on women [18]. There are also social and cultural norms that potentially influenced the reproductive health of both women and their children. For example, there is a perception that wives and children should be taken care of by the female family members in the postnatal period, hence partners are often not present [19]. In many situations, government policy also does not focus on male partner participation in maternal health services. In addition, gender inequality is a fundamental cause of women's constrained access to health services. For instance, decisions regarding mobility and expenditure for health care are mainly in the hands of men, which prevents women from seeking care for their health problems as well as MCHS utilization [19]. The existing evidence showed that

women who are under the influence of their male partners were 21% less likely to use PNC services than their counterparts [20]. Other studies also revealed that socio-demographic, individual, cultural, and health facility factors hinder the involvement of male partners in the PNC services [18, 21].

The International Conference on Population and Development (ICPD) has emphasized the need to "promote gender equality in all spheres of life and to promote men to take responsibility for their sexual and reproductive behavior and social and family roles" [22]. In 2015, the WHO recommended MNCH interventions to encourage men's involvement during pregnancy, childbirth, and the postpartum period to improve; home care practices for women and newborns, the use of skilled care, the timely use of facility care for obstetric and neonatal complications, and to augment couple communication mainly on family planning and contraceptive use [23]. Couple-friendly reproductive health services and male partner involvement in women's reproductive health have attracted substantial attention in recent years [24].

Even though the WHO [23], the 1994 ICPD [22], and the 1995 International Conference on Women in Beijing recommended male partner involvement in maternal health issues, in many SSA countries, including Ethiopia, maternal and child health is believed to be a woman's affair [25]. To improve this deep-rooted challenge, Ethiopia has made major efforts through the health extension program and the health development army. However, according to the 2015 MOH annual report, male partner involvement still remains a big challenge for the country and much needs to be done to overcome the problem [26]. Although few published studies are available from some regions of Ethiopia, they were primarily focusing on ANC, PMTCT, and BPCR [9, 19, 27]. It is essential to assess the extent of male partner's involvement in PNC service utilization to plan an effective intervention strategy to improve their involvement and improve health outcomes. Therefore, the current study aimed to assess male partner involvement in PNC service utilization and associated factors in Motta district, North West Ethiopia.

## Methods

### Study design and settings

A community-based cross-sectional study was conducted in Motta district, Northwest Ethiopia from March 16 to 30, 2020. In total, the district comprises 36 kebeles, of which 30 of them were rural and the remaining six were urban, with an estimated population of 189,714, of which 94,436 were males.

The eligible study participants were male partners whose wives gave birth in the last 12 months in selected kebeles and who lived in the study area for at least six months. Male partners who were critically ill during the data collection period and unable to communicate were excluded from this study. A two-stage random sampling method was adopted to draw sample respondents. In the first stage, 12 kebeles were randomly selected from a total 36 kebeles. Two kebeles were from urban areas and ten kebeles were from rural areas proportionally. In the second stage, out of the estimated 1857 male partners living in the selected kebeles, 612 eligible participants were recruited for this study by using systematic random sampling. The starting point was determined by lottery method and subsequent households were selected every k (3) interval.

### Sample size determination

The sample size for the first objective was calculated by using a single population proportion formula by taking a 5% margin of error, a 95% confidence interval, and the magnitude of male partner involvement was 59.3% from the study conducted in Tanzania [28], with a design effect of 1.5 and a 10% non-response rate. The sample size for the second objective was

determined using the double population formula by using Epi Info version 7 considering the following assumptions: a confidence interval (CI) of 95%, a 1.5 design effect, power of 80%, a ratio of 1:1, and a non-response rate of 10%. The factors were taken from a previous study conducted in Tanzania [28]. The largest sample size for this study was calculated from the first objective, which was 612.

## Data collection and management

A self-administered questionnaire was adopted after reviewing different kinds of literature [24, 28]. The questionnaire has six sections: The first section concerns the socio-demographic characteristics of the participants. The second section focuses on the assessments of participant's knowledge of PNC services and danger signs during the postpartum period. The third section assesses participant's attitudes toward PNC services. The fourth section assesses health service-related factors and the final section assesses participant's involvement in PNC service utilization. The data collection process took two weeks (from March 16 to March 30, 2020). Data was collected by five diploma midwives and three diploma nurses and supervised by two BSc midwives. The questionnaire was originally written in English, and then translated to the local language (Amharic) and back to English to ensure that the translated version gives the proper meaning. To maintain the quality of the data, the questionnaire was pretested on 5% of the sample size before the real data collection. Before collecting the data, training was given to the data collectors to familiarize them with the objective of the study and understand the survey questions. Once data had been collected, each questionnaire was checked daily for its completeness by the supervisors and researcher, and feedback was given to data collectors accordingly.

## Statistical analysis

The collected data was entered and cleaned using Epi Data version 3.1 and then exported to the Statistical Package for the Social Sciences (SPSS) version 23 for analysis. The data was summarized using descriptive analysis, and the outcome of the study was presented in the form of text, figures, and tables. A binary logistic regression analysis was carried out to see the association between independent and dependent variables. All explanatory variables with p<0.2 in bivariate analysis were included in the multivariate analysis and a significant association was identified based on p<0.05 and odds ratio with 95% CI in multivariable logistic regression. The final model fitness was checked by using the Hosmer-Lemeshow Goodness of Fit test (p = 0.156). Multicollinearity was checked using Variance inflation factor (VIF), which was less than 10. A wealth index was obtained using Principal Component Analysis (PCA). In the first analysis, components with eigenvalues (variance) greater than one were extracted. According to "Kaisers rule", only those components with eigenvalues greater than one should be retained [29]. Based on Kaiser's rule, the study decided to retain the first component because it had a greater eigenvalue (variance) than the other components. In the first component, the variables that had a correlation coefficient score of less than 0.3 were excluded in the second analysis [30]. Finally, two components with eigenvalues greater than one have been extracted. Based on the same "Kaisers rule", the first component was retained because it had a greater eigenvalue than the second component, and this first component was the one used to determine the participant's wealth index score.

## Measurement

In this study, a male partner is defined as a man who married a woman and was responsible for the pregnancy of that woman. Male involvement is a composite variable without a single

standard measurement scale. It was assessed using a ten-point index which includes: discussing with health professionals on PNC service and complications occurring in the postpartum period, discussing with a spouse about PNC service, accompanying spouse for PNC service, physical support, discuss postpartum contraception, emotional support, sharing decision-making power on PNC service with the spouse, financial support, help in domestic activities and looking after children [28, 31]. Each of these ten elements was given a score of (1) when the participant performed the activity and (0) when the activity was not performed. A total score was calculated and 50% was used as a cut-off point to classify participants as either involved (those who scored 50% and above from the total score) or not involved (those who scored less than 50% from the total score).

**Male partner involved in PNC service.** Those who had scored 50% or higher from the total score.

**Male partner accompanied in PNC service: A male partner who accompanied his spouse to the PNC clinic on at least one PNC visit, excluding the first 6-hour PNC visit.** *Knowledge about PNC service.* The knowledge of respondents about PNC service was evaluated using five questions. Each of these questions was given a score of (1) when answered correctly and (0) when the answer is incorrect. A total score was computed for each participant, and knowledge about PNC services was estimated using the mean score [32]. Male partners who scored mean or above were considered to have good knowledge about PNC services, whereas those who scored below the mean score were classified as having poor knowledge about PNC services.

*Good knowledge of danger signs.* those male partners who had replied three or more maternal or neonatal danger signs or complications [33].

*Attitude towards PNC service.* The attitude of the respondents towards PNC service was evaluated by using seven questions. The questions were designed using a Likert scale format with five answer alternatives; ranged from strongly agree, agree, indifferent, disagree, and strongly disagree. The overall score was measured, and respondents who scored the mean score or above were considered to have a favorable attitude towards PNC services, whereas those who scored below the mean score were classified as having an unfavorable attitude [32].

## Ethical considerations

The ethical committee of Bahir Dar University College of Medicine and Health Sciences approved this study. Further approval was also granted from the Motta district and each kebele administration. Confidentiality was assured and written informed consent was taken from all participants. The obtained information was kept anonymous and recorded in such a way that the respondent could never be known.

## Result

### Socio-demographic characteristics of participants and their wives

Among the total of 612 male partners, 595 male partners participated in the study making a response rate of (97%). The mean age was 41.8 with ±8.7 SD years and (48.2%) were in the age group of 35–44 years. Of 595 respondents, (72.9%) of the respondents were rural dwellers. More than half of the respondents did not attend formal education (59.2%) (Table 1).

### Knowledge of PNC service and danger signs during the PNC period

Of 595 respondents, (32.3%) had good knowledge of the PNC service. Regarding the source of information on PNC (27.1%) of respondents got information from their family, (35.1%) from

**Table 1. Socio-demographic characteristics of male partners and their wives in Motta district, North West Ethiopia, 2020 (n = 595).**

| Variables | Frequency | Percentage |
|---|---|---|
| **Age** | | |
| 25–34 | 103 | 17.3 |
| 35–44 | 287 | 48.2 |
| >44 | 205 | 34.5 |
| **Religion** | | |
| Orthodox | 507 | 85.3 |
| Muslim | 83 | 13.9 |
| Others* | 5 | 0.8 |
| **Residence** | | |
| Urban | 162 | 27.2 |
| Rural | 433 | 72.8 |
| **Ethnicity** | | |
| Amhara | 593 | 99.7 |
| Others** | 2 | 0.3 |
| **Husband education** | | |
| No formal education | 353 | 59.3 |
| Primary (1–8) | 71 | 11.9 |
| Secondary (9–12) | 81 | 13.7 |
| College and above | 90 | 15.1 |
| **Husband occupation** | | |
| Farmer | 427 | 71.8 |
| Merchant | 86 | 14.5 |
| Governmental employee | 61 | 10.2 |
| Other*** | 21 | 3.5 |
| **Mother education** | | |
| No formal education | 404 | 67.8 |
| Primary (1–8) | 101 | 17 |
| Secondary (9–12) | 36 | 6.1 |
| College and above | 54 | 9.1 |
| **Spouse occupation** | | |
| Housewife | 389 | 65.4 |
| Merchant/private | 126 | 21.2 |
| Governmental employee | 50 | 8.4 |
| Other*** | 30 | 5 |
| **No of children** | | |
| 1 | 96 | 16.1 |
| 2 | 99 | 16.6 |
| ≥3 | 400 | 67.3 |
| **Income** | | |
| Poor | 215 | 36.1 |
| Medium | 248 | 41.7 |
| Rich | 132 | 22.2 |

* = protestant, catholic

** = Tigray, Oromo

*** = student, daily laborer.

HEW, (21%) from health professionals, and (26.7%) from other sources like friends, relatives, media, books, and magazines. Around two-thirds (62%) of the respondents mentioned immunization as one of the PNC services to be provided during the PNC visit.

Regarding danger signs, about 243 (40.8%) of the respondents replied three or more danger signs and complications that could happen during the postpartum period. Of which, (49.4%) of the respondents stated vaginal bleeding (Table 2).

## Attitude towards PNC service

The overall male partner's favorable attitude towards PNC service was 25.5% (Table 3).

## Culture related factors

Of 595 respondents, (15.8%) perceived that PNC is a women's affair and that (15.5%) considered the PNC clinic as a place only for women. Nearly, one-third (27.2%) of the respondents had a culture of discussion with their partner about PNC service utilization. On the other hand, (20.3%) of respondents reported that some misconceptions and myths influence the involvement of the male partners in PNC service utilization.

## Health service-related characteristics

Of 595 respondents, (42.4%) had health facility (PNC clinic) access within 30 minutes. Similarly, (35.8%) of respondents had transport access to the PNC clinic. About (14.1%) of the respondents had good welcoming or appreciation by the health professionals when they visited

**Table 2. Male partners knowledge of danger signs and complications during the postpartum period in Motta district, North West Ethiopia, 2020 (n = 595).**

| Danger sign | Frequency | Percentage |
|---|---|---|
| **Maternal** | | |
| Fever | 116 | 19.5 |
| Severe headache | 145 | 24.4 |
| Heavy vaginal bleeding | 294 | 49.4 |
| Foul smelling vaginal discharge | 84 | 14.1 |
| Blurring of vision | 52 | 8.7 |
| Depression | 3 | 0.5 |
| Calf muscle pain, redness and swelling | 6 | 1.0 |
| **Newborn** | | |
| Fever | 114 | 19.2 |
| Cold during | 95 | 16 |
| Abnormal breathing | 119 | 20 |
| Abnormal body movement | 4 | 0.7 |
| Cord bleeding, redness, puss | 84 | 14.1 |
| Abnormal breastfeeding | 123 | 20.7 |
| Yellow discoloration of eye, palm and sole | 6 | 1.0 |
| **Complication** | | |
| Pregnancy-induced hypertension | 59 | 9.9 |
| Uterine/breast infection | 12 | 2 |
| PPH | 129 | 20.7 |
| DVT | 5 | 0.8 |
| Postpartum depression | 3 | 0.5 |
| Wound infection | 5 | 0.8 |

Table 3. Attitude of male partners towards PNC service in Motta District, North West Ethiopia, 2020 (n = 595).

| Variables | Frequency (percentage) | | | | |
|---|---|---|---|---|---|
| | Strongly agree | Agree | Neutral | Disagree | Strongly disagree |
| It is the waste of time for male partners to participate with their wives during PNC visit | 11 (1.8) | 72 (12.1) | 225 (37.8) | 252 (42.4) | 35 (5.9) |
| Male partners should give emotional, financial, and physical support to their wives during the PNC period | 241 (40.5) | 253 (42.5) | 84 (14.1) | 6 (1) | 11 (1.8) |
| Male partners must focus on job responsibilities during the postnatal period rather than to involve in caring for their wives and child | 10 (1.7) | 75 (12.6) | 214 (36) | 276 (46.4) | 20 (3.4) |
| PNC services should only be left to women alone | 16 (2.7) | 78 (13.1) | 222 (37.3) | 247 (41.5) | 32 (5.4) |
| Visiting health institutions during the postnatal period is important for mothers and their children. | 288 (48.4) | 183 (30.5) | 101 (17) | 2 (0.3) | 21 (3.5) |
| Recommend other males to involve in PNC service | 76 (12.8) | 90 (15.1) | 322 (54.1) | 11 (1.8) | 96 (16.1) |
| Mothers should follow PNC follow up within 42 days | 252 (42.4) | 190 (31.9) | 130 (21.8) | 1 (0.2) | 22 (3.7) |

the PNC clinic with their partner for PNC service, and (14.7%) of the respondents reported that there was privacy when the service was provided. Regarding waiting time, (27.3%) of the respondents received the service within one hour.

## Male partner involvement in PNC service utilization

The overall magnitude of male involvement in PNC service utilization was 124 (20.8%) with a 95% CI (17.6%, 24.1%). About (18.7%) of the respondents accompanied their partner to the health facility for PNC service utilization. Of those male partners who did not accompany their partner, being preoccupied with work, a belief that the PNC is an issue for women only, and a lack of knowledge that postpartum could result in different complications were the most cited reasons for not doing so (Table 4).

## Factors associated with male partner involvement in PNC service utilization

In the bivariate logistic regression analysis with (p-value< 0.2) age, residence, husbands education, spouses occupation, number of children, income, knowledge of PNC service, knowledge of danger signs and complications, attitude towards PNC service, and distance were candidate variables for multivariable regression. Further analysis by using multivariable logistic regression demonstrates that residence, male partner educational status, knowledge of PNC service, knowledge of danger signs, attitude towards PNC service, distance to the health facility, and waiting time to get the service were significantly associated with male partner involvement in PNC service utilization at p- value< 0.05.

The results in Table 5 indicated that male partners living in urban areas were 2.5 times [AOR = 2.5, 95%CI = (1.3, 4.8)] more likely to be involved in PNC service utilization than their counterparts. In addition, male partners with primary and secondary education were 2.5 and 2.2 times [AOR = 2.5, 95%CI = (1.5, 5.2)] and [AOR = 2.2, 95%CI = (1.0, 4.8)] more likely to be involved in PNC service utilization, respectively, as compared to partners with no formal education. The result also suggested that male partners who had good knowledge of PNC service were 2.5 times [AOR = 2.5, 95%CI = (1.5, 4.2)] more likely to be involved in PNC service utilization as compared to those who had poor knowledge of the service. Moreover, male

**Table 4. Male partner involvement in PNC service utilization in Motta district, North West Ethiopia, 2020(n = 595).**

| Components | | Frequency | Percentage |
|---|---|---|---|
| Discusses on PNC service with their partner | Yes | 175 | 29.6 |
| | No | 420 | 70.4 |
| Shared decision-making powers on PNC with wife | Yes | 186 | 31.3 |
| | No | 409 | 68.7 |
| Accompanies partner to the health care facility | Yes | 111 | 18.7 |
| | No | 484 | 81.3 |
| Discusses on family planning with their partner | Yes | 198 | 33.3 |
| | No | 397 | 66.7 |
| Discusses on PNC service and complications during the postpartum period with her health care provider | Yes | 68 | 11.4 |
| | No | 527 | 88.6 |
| Provides physical support to his partner during postnatal period | Yes | 68 | 11.4 |
| | No | 527 | 88.6 |
| Provides emotional support to their partner (encourage) for PNC service utilization | Yes | 182 | 30.6 |
| | No | 413 | 69.4 |
| Provides financial support to their partner for PNC service utilization | Yes | 214 | 36 |
| | No | 381 | 64 |
| Helps domestic tasks | Yes | 197 | 33.1 |
| | No | 398 | 66.9 |
| Looks after children | Yes | 176 | 29.6 |
| | No | 419 | 70.4 |

partners who had good knowledge of danger signs and complications that occur during the postpartum period were 2.5 times [AOR = 2.5, 95%CI = (1.4, 4.3)] more likely to be involved in PNC service utilization than their counterparts.

Male partners who had a favorable attitude toward PNC service were 2.6 times [AOR = 2.6, 95%CI = (1.6, 4.4)] more likely to be involved in PNC service utilization than their counterparts. A male partners involvement in PNC service was found to be 2.1 times [AOR = 2.1, 95% CI = (1.2, 3.9)] more likely among male partners who have health facility (PNC clinic) access within 30 minutes as compared to those respondents who have no health facility access within 30 minutes (Table 5).

## Discussion

The overall magnitude of male partner involvement in PNC service utilization was 20.8% with 95%CI (17.6%, 24.1%). The finding of this study was consistent with the study conducted in western Ghana, which was 20% [34]. This might be due to similar educational status. However, the finding of this study was lower than other studies conducted in Tanzania which was 59.3% [28], in Nigeria 87.7% [35], and in Nepal 33.8% [36]. This could be due to differences in socio-demographic status and study area.

This study revealed that residence was significantly associated with male partner's involvement in PNC service utilization. Male partners who resided in urban areas were 2.5 times more likely to be involved in PNC service utilization than those who resided in rural areas which is supported by the study conducted in Kenya [37]. One reason for this might be male partners who resided in urban areas might have a higher level of educational knowledge and exposure to information from different sources on PNC. The other justification might be related to the accessibility of public and private health care facilities and transportation in

**Table 5. Bivariable and multivariable logistic regression of factors associated with male partner involvement in PNC service utilization in Motta district, North West Ethiopia, 2020 (n = 595).**

| Variables | Male partner involvement (95%CI) | | |
|---|---|---|---|
| | COR | AOR | P-value |
| Age(years) | | | |
| 25–34 | 1 | 1 | |
| 35–44 | 0.6(0.4–1.1) | 0.9(0.4,2.1) | 0.840 |
| >44 | 0.6(0.3–1.0) | 1.1(0.4,2.8) | 0.832 |
| Residence | | | |
| Urban | 9.1(5.8–14.1) | 2.5(1.3,4.8) | 0.005** |
| Rural | 1 | 1 | |
| Husband education | | | |
| No formal education | 1 | 1 | |
| Primary (1–8) | 5.8(3.1, 10.9) | 2.5(1.2,5.2) | 0.015* |
| Secondary (9–12) | 8.3(4.6, 15.0) | 2.2(1.0,4.8) | 0.005** |
| College and above | 10.1(5.7, 17.9) | 1.5(0.7,3.4) | 0.328 |
| No of children | | | |
| 1 | 2.5(1.5, 4.1) | 1.4(0.7,2.9) | 0.330 |
| 2 | 2.0(1.2, 3.4) | 0.9(0.5,1.9) | 0.981 |
| ≥3 | 1 | 1 | |
| Wealth Index | | | |
| Poor | 1 | 1 | |
| Medium | 1.8(1.1, 3.0) | 1.6(0.9,2.8) | 0.144 |
| Rich | 2.4(1.4, 4.1) | 1.6(0.8,3.1) | 0.169 |
| Knowledge of PNC service | | | |
| Poor | 1 | 1 | |
| Good | 7.4(4.8, 11.5) | 2.5(1.5,4.2) | 0.001** |
| Knowledge of danger signs during PNC | | | |
| Poor | 1 | 1 | |
| Good | 6.8(4.3, 10.7) | 2.5(1.4,4.3) | 0.001** |
| Attitude to the service | | | |
| Unfavorable | 1 | 1 | |
| Favorable | 2.9(1.9, 4.5) | 2.6(1.6,4.4) | 0.000*** |
| Distance from PNC service | | | |
| >30 minutes | 1 | 1 | |
| ≤30 minutes | 9.9(4.4, 11.0) | 2.1(1.2,3.9) | 0.016* |

*p-value<0.05,

**p-value<0.01,

***p-value<0.001.

urban areas [38]. However, the finding of this study was inconsistent with a study done in Tanzania [28], which showed that urban residents were two times less likely to be involved in PNC service utilization.

Male partners who had primary and secondary educational status were 2.5 times and 2.2 times more likely to be involved in PNC service utilization as compared to those who had no formal education, respectively. The finding was supported by a study done in Myanmar and Nepal [36, 39]. This may be because education enables a better understanding of the complications and danger signs that occur during the postnatal period as well as the importance of

PNC service, which could encourage them to participate in PNC service utilization [19]. In addition, education improves discussions between husband and wife about health-related issues and other topics [36]. Furthermore, education also allows the male partners to reject the negative cultural beliefs, and it is also likely that educated male partners have some formal employment that enables them to raise funds that they can use to get postnatal care services like transportation [20].

It is also found that male partners who had good knowledge of PNC service utilization were 2.5 times more likely to be involved in PNC service utilization than their counterparts. This result is supported by a study done in Tanzania and Nepal [28, 36]. This suggests that having awareness about the significance of PNC service may lead male partners to make informed decisions and consequently, result in positive behavioral change, which could encourage them to become involved in PNC services with their partners [28]. Moreover, having good knowledge of PNC services may improve their attitude towards the services and increase their motivation to consistently support and encourage their partners to use PNC services.

The finding also showed that male partners with good knowledge of danger signs were 2.5 times more likely to be involved in PNC service utilization than their counterparts, which is consistent with other studies done in Bangladesh and Uganda [33, 40]. This might be because knowledge of danger signs leads to greater anticipation to lessen the effects of postpartum complications by reducing the first two delays (delay in decision-making and delay in transportation) [19]. The other possible explanation for this might be that awareness of danger signs during the postpartum period is an important factor in motivating male partners and their families to attend health care services at the earliest opportunity with the intention of prevention and early detection [41].

The study also showed that male partners who had a favorable attitude towards PNC service were 2.6 times more likely to be involved in PNC service utilization than those with unfavorable attitudes. This finding was supported by a study conducted in Zambia [42]. This might be because men's positive attitudes enhance maternal health care utilization because of the increased participation of their partners [43]. Furthermore, when male partners believe their primary responsibility is to secure financial issues but that activities such as accompanying spouses to health care facilities should be reserved for women, their involvement becomes decreased; however, when they believe the opposite, their involvement increases [21].

This study established a statistically significant positive relationship between distance and male partner involvement in PNC service utilization. Male partners who had health facility access within 30 minutes were 2.1 times more likely to be involved in PNC service utilization than their counterparts which correspond with other similar studies in Zambia [42] and Uganda [40]. This might be because the long distance to the health facilities is one of the most important determinants in the decision not to seek modern health care even when needed and there is no access to transport or an increase in transportation cost [44]. Furthermore, due to increasing distance from health facilities, there is an increase in lost production time as well as a possible lower exposure to health -related information as a result male partners involvement becomes decreased [45].

Previous studies mostly focused on male partner involvement in ANC, BPCR, or institutional delivery; however, the magnitude and associated factors of the male partner in PNC services are not clear. Therefore, this study assesses the impact of the male partner's involvement on the female partner's use of PNC services. Our findings showed that the active participation of men appeared to have more impact on women's PNC service utilization. This study also reinforces the findings on the significant influence of education, knowledge, and attitude on PNC services and health facility- related factors on male partner involvement in PNC service utilization. The impact of a male partners involvement on maternal and childbirth outcomes,

how men could have better involvement and the time of involvement (during pregnancy, delivery, and postpartum period) merit further investigations.

This study indicated a significant positive impact of male involvement on maternal health through improved utilization of PNC services. It implies the need to shift from women-only maternal health services to male-friendly or couple-friendly services and also to improve health-care/government policies that isolate men from having active engagement in maternal health.

Since this study was undertaken at the community level, it is representative of the source population. The probability sampling method used increases the representativeness of the study population. However, one of the limitations of this study is that variables like knowledge of PNC service, knowledge of danger signs during PNC, and attitude to the service are measured at the interview time, not prior to childbirth and, as a result, can be affected by issues of reverse causality. The cross-sectional nature of this study does not address the underlining social- cultural gender norms, which may limit the male partner's involvement in PNC. In addition, respondents may also be exposed to social desirable bias and recall bias.

## Conclusions

This study examined male partner involvement in postnatal care service utilization and factors determining their participation in Motta district, Northwest Ethiopia. In general, a male partner's involvement in their spouses PNC service utilization was found to be low. The regression result also showed that male partner involvement was positively and significantly associated with place of residence, educational status, knowledge of PNC service utilization, knowledge of danger signs, attitude toward PNC service and distance to health facility. Findings from this study suggested that policies aimed at improving awareness in relation to PNC service, expanding health service facilities in the nearby areas, promoting educational attainment, and enhancing the efficiency of health service provisions could be relevant mechanisms used to improve male partner's involvement in PNC service. Community-based awareness creation through various channels such as radio, television, magazines, and campaigning programs could help to increase the male partner's participation in PNC services. It is also suggested that further qualitative studies may be important in order to address socio- cultural gender norms.

## Supporting information

**S1 Appendix. Survey questionnaire.**
(DOCX)

**S1 File. Description of PC analysis.**
(DOCX)

**S1 Dataset.**
(SAV)

## Acknowledgments

The author acknowledges the support of supervisors, and data collectors. The author is also very thankful to all the study participants.

## Author Contributions

**Conceptualization:** Alemwork Abie, Getahun Belay, Eden Asmare.

**Investigation:** Getahun Belay.

**Methodology:** Alemwork Abie, Nigusu Ayalew, Nigus Kassie.

**Software:** Alemwork Abie, Nigus Kassie.

**Supervision:** Eden Asmare.

**Writing – original draft:** Alemwork Abie, Eden Asmare, Nigusu Ayalew.

**Writing – review & editing:** Wondu Feyisa.

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
