## [Decision Letter · Decision Letter 0]

1 Oct 2021

PONE-D-21-03079Male partner involvement in postnatal care service utilization and associated factors: A community based cross sectional study in Motta district, Northwest EthiopiaPLOS ONE

Dear Dr. Getu,

Thank you for submitting your manuscript to PLOS ONE. After careful consideration, we feel that it has merit but does not fully meet PLOS ONE’s publication criteria as it currently stands. Therefore, we invite you to submit a revised version of the manuscript that addresses the points raised during the review process.

We look forward to receiving your revised manuscript.

Kind regards,

Zelalem Nigussie Azene, MPH

Academic Editor

PLOS ONE

Additional Editor Comments (if provided):

This is an important study, as it offers a benchmark evidence-base of the male involvement in postnatal care service utilization and its determinants in Motta district, Ethiopia

The quality of the English used throughout your manuscript does not currently meet our requirements, as there are several incorrect sentence constructions and grammatical errors throughout obscuring the message the authors want to convey. We recommend that you ask a native English speaking colleague to help you copy-edit the paper. If this is not possible, you may need to use a professional language editing service. Use of an editing service is neither a requirement nor a guarantee of acceptance for publication.Can you please review all statistical reporting in your manuscript? We would encourage you to seek additional advice from a statistician before submitting your revised paper. It is important that the statistical analyses used are suitable for your study and that all statistical reporting is correct. Can you please confirm in your cover letter whether you have addressed this point?       I request the authors do add the implications of this particular study at the end of the discussion section.Please, include the added value of this work at the end of the discussion section.In Abstract section, the authors state “favorable attitude [AOR=2.61, 95%CI= (1.55, 4.39)], short distance [(AOR=2.35, 95%CI=1.30, 4.26)]”, however, favorable attitude?? And short distance?? These phrases are incomplete that can potentially baffle the reader. Please, do them complete by stating the full variable (favorable attitude towards-------) this issue is also happened in operational definition section.Please, see again the operational definition section, there even variables, which actually don’t need operationalization but did.Very amazingly, there are a huge overlapping of words even sentences from the so far works, please, avoid this.“Conclusion and recommendation”: Please, do erase recommendation and rewrite as conclusions. And also “Community-based awareness creation through health education should be provided to raise their involvement, knowledge, and attitude on postnatal care service”, these should only be the ways forwarded to tackle the problem studied.“PNC service, Male involvement, Ethiopia”: Are these only keywords?  

 I, therefore, request the authors to make major revisions before resubmitting their manuscript.

Journal Requirements:

2. Thank you for stating in the text of your manuscript "The ethical committee of Bahir Dar University College of Medicine and Health sciences approved this study. Further approval was also granted from Motta district and each kebeles administration. Informed consent from all participants was obtained before conducting this study. The obtained information was kept anonymous and recorded in such a way that the respondent could never be known". 

Please specify what type of consent you obtained (for instance, written or verbal, and if verbal, how it was documented and witnessed).

Please also add all of this information to your ethics statement in the online submission form.

3. Please include additional information regarding the survey or questionnaire used in the study and ensure that you have provided sufficient details that others could replicate the analyses. For instance, if you developed the survey or questionnaire as part of this study and it is not under a copyright more restrictive than CC-BY, please include a copy, in both the original language and English, as Supporting Information. If the questionnaire is published, please provide a citation to the (1) questionnaire and/or (2) original publication associated with the questionnaire.

4. We suggest you thoroughly copyedit your manuscript for language usage, spelling, and grammar. If you do not know anyone who can help you do this, you may wish to consider employing a professional scientific editing service. 

Reviewers' comments:

Reviewer's Responses to Questions

**Comments to the Author**

1. Is the manuscript technically sound, and do the data support the conclusions?

Reviewer #1: Yes

Reviewer #2: Partly

2. Has the statistical analysis been performed appropriately and rigorously? 

Reviewer #1: Yes

Reviewer #2: Yes

3. Have the authors made all data underlying the findings in their manuscript fully available?

Reviewer #1: Yes

Reviewer #2: Yes

4. Is the manuscript presented in an intelligible fashion and written in standard English?

Reviewer #1: Yes

Reviewer #2: No

5. Review Comments to the Author

Reviewer #1: The authors selected an important topic of time for The study site and related context. In their Introduction an additional information regarding traditional gender roles expected of men and women in Ethiopia would benefit the paper in terms of measurements and hence outcomes /results. Authors may look at this point and discuss the point in the discussion section to aid better recommendations

Ethical considerations paragraph may be improved to include confidentiality and issues of anonymity.

The authors may want to write briefly about limitations and straight of their study approach/methodological considerations at the beginning of the Discussion section to improve rigour.

Few typos may be looked at again

The paper is generally good but I suggest you rectify my few suggests prior publication in PLOS ONE

Reviewer #2: This is an interesting topic, but unfortunately, the paper needs quite some more work. The language shows that the authors are not native speakers nor very experienced in academic write up in English, so an language editor would be good. The sample size seems sufficient, but the analysis is very descriptive for the most. I miss a better cultural and social description on gender issues in pregnancy in Ethiopia. I miss a more thorough explanation of the state of the art in Post partum care in general. The findings are interesting and pretty detailed, but a numeric comparison between countries make not so much sense. The situation in Ethiopia is interesting enough in its own right. So why, or why not, are men there for the women?

Abstract

Do not include too much detail in the abstract such as statistical method, confidence intervals or likewise. Start with a small problem statement; “so what is at stake here”….. An abstract should inspire readers to grasp the idea and to look for more detail in the text. An abstract should capture the essence of the only.

Article. I would start by describing what male participation actually IS. Most over the industrialized world has witnessed an ever increasing presence of partner during birth , in antenatal care visits, as part-takers in maternity and paternity leave from work, etc. These developments have not always come on as projects pr policy, but evolved from increased gender sensitivity and family choice. The article should describe the state of the art in delivery services in Ethiopia (many deliver at home, partners are often not present, neonatal mortality is still high etc:

I do not really see how male participation in general is linked to better mother and child survival, this has to be explained (better access to resources, faster transport? Better negotiating power?) So the whole introductory part needs to be re written, and don’t use statements that are not documented by evidence.

In methods. The statement “reviewing different literatures” needs references, Which studies did you learn from?

Results. No need to repeat table results in text.Shorten the presentation of results a little.

Chose one way to present

Please refrain from using too many decimals in estimations. One decimal is often enough.

I miss a mote thick discussions on ways forward, based on the findings.

6. PLOS authors have the option to publish the peer review history of their article (what does this mean?). If published, this will include your full peer review and any attached files.

Reviewer #1: No

Reviewer #2: No

---

## [Author Response · Author response to Decision Letter 0]

9 Dec 2021

Hi dear Editor and reviewers, we have carefully read and applied the suggestions from reviewers and comments from the editor as follow: 

Response to Editor

• “We suggest you thoroughly copyedit your manuscript for language usage, spelling, and grammar”, we have tried to correct the language problems as much as possible 

• “Can you please review all statistical reporting in your manuscript?” we have reviewed the statistical reporting and we have made some modifications

• “I request the authors do add the implications of this particular study at the end of the discussion section”, the implication and add value of this study is included at the end of the discussion section as you suggested.

• “In Abstract section, the authors state “favorable attitude [AOR=2.61, 95%CI= (1.55, 4.39)], short distance [(AOR=2.35, 95%CI=1.30, 4.26)]”, however, favorable attitude?? And short distance?? These phrases are incomplete that can potentially baffle the reader. Please, do them complete by stating the full variable (favorable attitude towards-------) this issue is also happened in operational definition section” we have tried to complete the incomplete phrases in the abstract and operational definition section

• “Please, see again the operational definition section, there even variables, which actually don’t need operationalization but did”, we have revised this section and tried to avoid variables which don’t need operationalization

• “Please, do erase recommendation and rewrite as conclusions” we have erased the word recommendation and merge the components with the conclusion section

• PNC service, Male involvement, Ethiopia: “Are these only keywords?” we have added other keywords

• We have checked multicolinearity between independent variables using VIF which was 6

• The obtained information was kept anonymous and recorded in such a way that the respondent could never be known. “Please specify what type of consent you obtained”, we have specified the type of consent that we used

• “please include a copy, in both the original language and English, as Supporting Information” we have addressed this issue

• Data set is also included in supporting information

Response to Reviewer #1

• “In their Introduction an additional information regarding traditional gender roles expected of men and women in Ethiopia would benefit the paper in terms of measurements and hence outcomes /results”, we have tried to add some information regarding traditional gender roles in the introduction section but since we have no finding or associated factor related with this issue, this point does not included in the discussion section

• “Ethical considerations paragraph may be improved to include confidentiality and issues of anonymity”, we have included issue of confidentiality and anonymity

• “Write briefly about limitations and strength of their study approach/methodological considerations at the beginning of the Discussion section”, but we do not want to put the strength and limitation of the study at the beginning of the discussion section rather we have put it at the end of the discussion 

Response to Reviewer #2

• “The language shows that the authors are not native speakers nor very experienced in academic write up in English”, we have tried to correct the language problems as much as possible

• “I miss a better cultural and social description on gender issues in pregnancy in Ethiopia. I miss a more thorough explanation of the state of the art in Postpartum care in general.”, we have tried to state cultural and social description of gender issue on postpartum care

• “A numeric comparison between countries make not so much sense”, we have deleted the numeric comparison between countries

• “Do not include too much detail in the abstract such as statistical method, confidence intervals or likewise”, we have avoided unnecessary details in the abstract section that you have mentioned

• “I do not really see how male participation in general is linked to better mother and child survival; this has to be explained (better access to resources, faster transport? Better negotiating power?)”, we have addressed this point by describing the link between male partner participation and better mother and child survival

• “The statement “reviewing different literatures” needs references, which studies did you learn from?”, we have put the references that we used for preparing the tool

• “No need to repeat table results in text. Shorten the presentation of results a little”, we have tried to shorten the presentation of results and avoided some repetitions

• “Please refrain from using too many decimals in estimations. One decimal is often enough”, we have tried to decrease decimals

Thanks for your constructive suggestions and comments

---

## [Decision Letter · Decision Letter 1]

20 Apr 2022

PONE-D-21-03079R1Male partner involvement in postnatal care service utilization and associated factors: A community based cross sectional study in Motta district, Northwest EthiopiaPLOS ONE

Dear Dr. getu,

Thank you for submitting your manuscript to PLOS ONE. After careful consideration, we feel that it has merit but does not fully meet PLOS ONE’s publication criteria as it currently stands. Therefore, we invite you to submit a revised version of the manuscript that addresses the points raised during the review process.

There has been a change of academic editor from the previous version. The manuscript was sent back to the previous reviewers. Only reviewer 1 was available. The new academic editor, based on a detailed revision of the revised manuscript and the review process is able to make a decision, noting that the readability is much improved but there are important issues in the formulation of the model, the reporting, and the interpretation of results that need to be addressed in order to meet PLOS ONE guidelines.

Sample size determination:

Two objectives (1 and 2) are mentioned on p.5-6, but it is not disclosed what these objectives are. Probably it should have been mentioned at the end of the introduction, that is too succinct regarding the goals of this particular study.

p. 8: “595 male partners were participated” Remove “were”

There is an apparent contradiction in the analysis. It is said that “**Male partner involved in PNC service: **those who had scored median value and above” (this would imply that, by definition, 50% are involved). Then: “The overall magnitude of male involvement in PNC service utilization was 124 (20.8%)”. Is the same name being used for two different things? Please clarify. You should also provide some description of the scale of involvement. Table 4 provides the item by item, but there is no information on the distribution of the scale of the 10-point index.

A more important caveat is the lack of justification for the inclusion of “knowledge of PNC service”, “Knowledge of danger signs during PNC”, “attitude to the service” as “independent variables”. Note that the data is compiled AFTER the childbirth. Therefore, it is not possible to talk about the effect of knowledge of PNC service prior to pregnancy (which would be the magnitude of interest) with this data. If, after birth, there is lack of knowledge it is a sign of low involvement, but the causality goes the other way round: because of low involvement the opportunity was lost to get to know PNC services during pregnancy. The suggestion is to REMOVE this variables from the analysis. Note also, that the inclusion probably is also making the socioeconomic variables less interpretable, eg: education: It would be involvement of partner with different education level for a given level of knowledge of PNC. You’d expect part of the effect of education is through better knowledge. If you control for this factors, the interpretation of the socioeconomic variables becomes very difficult.

An alternative: It would be defensible to extend the index of involvement and include as extra points in the dependent variable the knowledge of PNC service and danger signs AFTER the childbirth. Distance from PNC service can be used, although the limitation should be commented of being measured after the childbirth.

I am not so sure about the other service variables such as waiting time, privacy, and welcoming. If those variables are only available for those who attended some ANC service, they should not be used. If the are reported by everyone, how do they no? Instead of actual characteristics of the service it may refer to their preconceptions. My inclination is also to drop them from the analysis keeping only the clearly exogeneous variables

On a related topic: there is no talk about missing values and the “don’t know” responses. How were they treated?

Please also provide in the appendix the details regarding the PC analysis for constructing the wealth index and the final index used.

A section on limitations needs to be added. The major limitation is that the attitudes are observed AFTER and not before the childbirth.

We look forward to receiving your revised manuscript.

Kind regards,

José Antonio Ortega, Ph.D.

Academic Editor

PLOS ONE

Reviewers' comments:

Reviewer's Responses to Questions

**Comments to the Author**

1. If the authors have adequately addressed your comments raised in a previous round of review and you feel that this manuscript is now acceptable for publication, you may indicate that here to bypass the “Comments to the Author” section, enter your conflict of interest statement in the “Confidential to Editor” section, and submit your "Accept" recommendation.

Reviewer #1: All comments have been addressed

2. Is the manuscript technically sound, and do the data support the conclusions?

Reviewer #1: Yes

3. Has the statistical analysis been performed appropriately and rigorously? 

Reviewer #1: Yes

4. Have the authors made all data underlying the findings in their manuscript fully available?

Reviewer #1: Yes

5. Is the manuscript presented in an intelligible fashion and written in standard English?

Reviewer #1: Yes

6. Review Comments to the Author

Reviewer #1: The authors have fairly addressed the editors' comments prior given to them. However, I have one more comment for authors to address. The Authors need to write the limitations/methodological considerations of this study given that one of the main factors for male participation in maternal health services with their partners /wives is the underlining social cultural gender norms where a cross sectional study may be considered to have some limitations in exploration of group norms.

I should also suggest that in for future further qualitative studies may benefit the topic in Ethiopia and for now I do offer one reference if authors would wish strengthen their discussion section. Maendeleo B, Matovelo D, Laisser R, Swai H, Yohani V, Tinka S, et al (2021) Men's perspectives on attending antenatal care visits with their pregnant partners in Misungwi district Rural Tanzania; A qualitative study. BMC Pregnancy and Childbirth 21 (1),1-8.

7. PLOS authors have the option to publish the peer review history of their article (what does this mean?). If published, this will include your full peer review and any attached files.

Reviewer #1: No

---

## [Author Response · Author response to Decision Letter 1]

4 Jun 2022

Hello dear Editor and reviewer, we have carefully read and applied the suggestions from you as follows:

Response to Editor

• “Two objectives (1 and 2) are mentioned on p.5-6, but it is not disclosed what these objectives are. Probably it should have been mentioned at the end of the introduction, that is too succinct regarding the goals of this particular study”

- Since it has been stated at the end of the introduction section on page-5(‘therefore, the current study aimed to assess male partner involvement in PNC service utilization and associated factors in Motta district, North West Ethiopia’), we said ‘objective one and two’. But after reading your comment, we did some modifications. 

• Male partner involved in PNC service: those who had scored median value and above “(this would imply that, by definition, 50% are involved)”. Then: The overall magnitude of male involvement in PNC service utilization was 124 (20.8%). “Is the same name being used for two different things? Please clarify”. 

- Dear editor if we understand your question, it mean that 124(20.8%) of male partners scored ≥50% (median) from the given activities which measure the involvement of male partners.

• “You should also provide some description of the scale of involvement. Table 4 provides the item by item, but there is no information on the distribution of the scale of the 10-point index”. 

- We think that this information is addressed in the operational definition section that is ‘Each of ten points was given a score of (1) when the participant performed the activity and (0) when the activity was not performed. A total score was computed for each participant and median score (50%) was used as a cut-off point to categorize into involved (those scored≥ 50% from the given activities) or not involved (<50% from the given activities). Since the data shows skewed distribution, median value was used as a cut-off point’. 

• “A more important comment is the lack of justification for the inclusion of knowledge of PNC service, Knowledge of danger signs during PNC, attitude to the service as independent variables. Note that the data is compiled AFTER the childbirth….. because of low involvement the opportunity was lost to get to know PNC services during pregnancy”.

- Dear editor if we understand your question, we know that knowledge and attitude towards PNC service are the possible factors that affect male partner involvement in maternal health service as showed by different studies. Since those male partners with good knowledge may understand well the possible birth complications; so that they encourage their spouses to attend PNC service. As well as husbands’ positive perception on benefit of maternity care might lead to higher level of male partners’ participation. Generally this finding point to the important roles of male partner’s knowledge and attitude in influencing their involvement in postnatal care. Regarding to their opportunity to get knowledge and develop attitude towards PNC service, it could be through mass media, books, magazines or health education by Health extension workers, health care providers or another bodies. In the other way, this could be also the limitation of cross sectional study design to show cause and effect relationship.

• “Note also, that the inclusion probably is also making the socioeconomic variables less interpretable: eg: education: It would be involvement of partner with different education level for a given level of knowledge of PNC. You’d expect part of the effect of education is through better knowledge…..”

- If we understood your comment, even if there is confounding, we can filter out such like issues with double analysis or multiple regression analysis. 

- And based on our understanding those who were educated might not have good knowledge and also those who were uneducated might not have poor knowledge.

• “I am not so sure about the other service variables such as waiting time, privacy, and welcoming”.

- We have removed these variables from the analysis based on your valuable comment.

• “On a related topic: there is no talk about missing values and the “don’t know” responses. How were they treated?”

- The ‘don’t know’ responses were recoded with ‘No’ responses for analysis and missing values were dropped

- But we have already removed those variables (like waiting time, privacy, welcoming, etc) with missing value and don’t know responses

• “Please also provide in the appendix the details regarding the PC analysis for constructing the wealth index and the final index used”.

- The details of PCA description has found in the appendix

Response to Reviewer

• “The Authors need to write the limitations/methodological considerations of this study given that one of the main factors for male participation in maternal health services with their partners /wives are the underlining social cultural gender norms…..”

- Dear reviewer we have accepted your comment and put as a limitation of this study. In addition to this we have recommended for further qualitative studies.

- And we would like to say thank you for the reference that you suggested.

---

## [Editor Report · Decision Letter 2]

15 Jun 2022

PONE-D-21-03079R2Male partner involvement in postnatal care service utilization and associated factors: A community based cross sectional study in Motta district, Northwest EthiopiaPLOS ONE

Dear Dr. getu,

Thank you for submitting your manuscript to PLOS ONE. After careful consideration, we feel that it has merit but does not fully meet PLOS ONE’s publication criteria as it currently stands. Therefore, we invite you to submit a revised version of the manuscript that addresses the points raised during the review process.

The manuscript has improved but some of the previous comments have not been properly addressed.

Regarding a possible conflict in the use of the term “median”: Median is a technical term referring to the location of the 50% percentile of a variable. As I state, there is always by definition 50% above and 50% below for a given variable. You say that “Male partner involved in PNC service: those who had scored median value and above” and that they are only 20.8%. That is not a median. Do you mean those that scored at least half of the total score? Then say so.

You again refer to “median score (50%)” Replace all such instances accordingly. It is wrong. All the instances of the word “median” in your reply are wrong!

• “A more important comment is the lack of justification for the inclusion of knowledge of PNC service, Knowledge of danger signs during PNC, attitude to the service as independent variables. Note that the data is compiled AFTER the childbirth….. because of low involvement the opportunity was lost to get to know PNC services during pregnancy”.

- Dear editor if we understand your question, we know that knowledge and attitude towards PNC service are the possible factors that affect male partner involvement in maternal health service as showed by different studies. Since those male partners with good knowledge may understand well the possible birth complications; so that they encourage their spouses to attend PNC service. As well as husbands’ positive perception on benefit of maternity care might lead to higher level of male partners’ participation. Generally this finding point to the important roles of male partner’s knowledge and attitude in influencing their involvement in postnatal care. Regarding to their opportunity to get knowledge and develop attitude towards PNC service, it could be through mass media, books, magazines or health education by Health extension workers, health care providers or another bodies. In the other way, this could be also the limitation of cross sectional study design to show cause and effect relationship.

I am not saying that those factors are not important. I am saying that your measurement is weak because it takes place post-partum, and it might be the case that the knowledge of PNC happened during or after because, for instance, a difficult delivery. Or the attitude to the service might have changed due to the use (or problems using) ANC. You have to state as one of the limitations that this variables are measured at the interview time, not prior to childbirth, and as a result can be affected by issues of reverse causality. You should raised these concerns in the section on the limitations of the study.

“On a related topic: there is no talk about missing values and the “don’t know” responses. How were they treated?”

- The ‘don’t know’ responses were recoded with ‘No’ responses for analysis and missing values were dropped

- But we have already removed those variables (like waiting time, privacy, welcoming, etc) with missing value and don’t know responses

It would be important to state what proportion of the observations had missing data and have been dropped.

We look forward to receiving your revised manuscript.

Kind regards,

José Antonio Ortega, Ph.D.

Academic Editor

PLOS ONE
---

## [Author Response · Author response to Decision Letter 2]

30 Jul 2022

Hello dear Editor, we have carefully read and applied the suggestions from you as follows:

• “Regarding a possible conflict in the use of the term “median”: Median is a technical term referring to the location of the 50% percentile of a variable. Do you mean those that scored at least half of the total score? Then say so”.

- Yes, It means that those scored half and above from the total score.

• You again refer to “median score (50%)” Replace all such instances accordingly. It is wrong. All the instances of the word “median” in your reply are wrong! 

- Thank you for your suggestion; we have replaced all such instances based on your recommendation. Page number8- 9, line number 185-189.

• “A more important comment is the lack of justification for the inclusion of knowledge of PNC service, Knowledge of danger signs during PNC, attitude to the service as independent variables. Note that the data is compiled AFTER the childbirth….. because of low involvement the opportunity was lost to get to know PNC services during pregnancy”. “You have to state as one of the limitations that this variables are measured at the interview time, not prior to childbirth, and as a result can be affected by issues of reverse causality. You should raised these concerns in the section on the limitations of the study”.

- We have understood your concern and have accepted your suggestion to correct it in such a way. So that we have addressed this issue in the limitation part. Page number 19, line number 403-406.

• “It would be important to state what proportion of the observations had missing data and have been dropped”.

- Based on our finding those who accompanied their partner for PNC visit were 18.7% (Page number 12, line 283). Since the remaining 81.3% of the participants didn’t accompany their partner, they couldn’t answer these questions like waiting time, privacy and welcoming which mean 81.3% of observations had missing data so that these variables (waiting time, privacy, welcoming, etc) have been dropped from the regression analysis.

# Reviewer

• But we did not receive additional points raised by the reviewer.

---

## [Editor Report · Decision Letter 3]

19 Aug 2022

PONE-D-21-03079R3Male partner involvement in postnatal care service utilization and associated factors: A community based cross sectional study in Motta district, Northwest EthiopiaPLOS ONE

Dear Dr. getu,

Thank you for submitting your manuscript to PLOS ONE. After careful consideration, we feel that it has merit but does not fully meet PLOS ONE’s publication criteria as it currently stands. Therefore, we invite you to submit a revised version of the manuscript that addresses the points raised during the review process.

Thank you for responding to all the reviewers' requests in the previous rounds of review. There are no further reviewer comments to address. However, please note that there are still many language errors in your submission. PLOS ONE does not copy edit accepted manuscripts (https://journals.plos.org/plosone/s/criteria-for-publication#loc-5). To that effect, please ensure that your submission is free of typos and grammatical errors.

We look forward to receiving your revised manuscript.

Kind regards,

Steve Zimmerman, PhD

Associate Editor, PLOS ONE
---

## [Author Response · Author response to Decision Letter 3]

2 Oct 2022

Hello, dear Editor, we have carefully read and applied the comments from you as follows:

• “Please note that there are still many language errors in your submission”.

o Thank you for your comment, and we have tried to use all efforts to edit the language errors.

• “If you would like to make changes to your financial disclosure, please include your updated statement in your cover letter”.

o We have not changed our financial disclosure.

Journal Requirements: 

• “If you have cited papers that have been retracted, please include the rationale for doing so…”

o We have not used retracted papers.

• “Any changes to the reference list should be mentioned in the rebuttal letter that accompanies your revised manuscript”.

o We have not changed our reference lists.

---

## [Editor Report · Decision Letter 4]

10 Oct 2022

Male partner involvement in postnatal care service utilization and associated factors: a community based cross sectional study in Motta district, Northwest Ethiopia

PONE-D-21-03079R4

Dear Dr. Alemwork Abie Getu,

We’re pleased to inform you that your manuscript has been judged scientifically suitable for publication and will be formally accepted for publication once it meets all outstanding technical requirements.

Kind regards,

Yann Benetreau

Staff Editor

PLOS ONE

Additional Editor Comments (optional):

In your final version, please address the following issues:

* line 71: evidences -> evidence

* line 91: does not focused -> does not focus

* Please do not include funding sources in the Acknowledgments or anywhere else in the manuscript file. Funding information should only be entered in the financial disclosure section of the submission system. https://journals.plos.org/plosone/s/submission-guidelines#loc-acknowledgments
---

## [Editor Report · Acceptance letter]

12 Jan 2023

PONE-D-21-03079R4 

Male partner involvement in postnatal care service utilization and associated factors: a community-based cross-sectional study in Motta district, Northwest Ethiopia. 

Dear Dr. Abie:

I'm pleased to inform you that your manuscript has been deemed suitable for publication in PLOS ONE. Congratulations! Your manuscript is now with our production department. 

Kind regards, 

on behalf of

Dr Steve Zimmerman 

Staff Editor

PLOS ONE